# Opinions and Knowledge of Parents Regarding Preventive Vaccinations of Children and Causes of Reluctance toward Preventive Vaccinations

**DOI:** 10.3390/ijerph17103694

**Published:** 2020-05-24

**Authors:** Anna Lewandowska, Tomasz Lewandowski, Grzegorz Rudzki, Sławomir Rudzki, Barbara Laskowska

**Affiliations:** 1Institute of Healthcare, State School of Technology and Economics, 37-500 Jaroslaw, Poland; barbara.laskowska917@gmail.com; 2Institute of Technical Engineering, State School of Technology and Economics, 37-500 Jaroslaw, Poland; tom_lew@interia.pl; 3Chair and Department of Endocrinology, Medical University of Lublin, 20-059 Lublin, Poland; grzegorz.rudzki@orange.pl; 4Chair and Department of General and Transplant Surgery and Nutritional Treatment, Medical University of Lublin, 20-059 Lublin, Poland; slawomir.rudzki@umlub.pl

**Keywords:** vaccinations, vaccination program, infectious diseases, parents’ attitude

## Abstract

*Background*: Despite the stability of global vaccination coverage, over 19 million children worldwide do not currently receive basic vaccines. Over the past several years, there has been a dramatic drop in the number of vaccinated children worldwide. The implementation of the vaccination program and the scope of protection depend on the parents or legal guardians, who decide whether to vaccinate their child or not. Studies were conducted to assess parents’ knowledge, attitudes, and beliefs about vaccines, as well as the role of healthcare providers in parents’ decisions. *Methods*: A population survey was conducted in 2018–2019. Parents or legal guardians of the children were invited to participate in the study during their visits to the clinic for healthy or sick children. The method used in the research was a diagnostic survey. *Results*: According to the conducted research, men and women constituted 45% and 55% of participants, respectively. The average age of men was 44, while, for women, it was 41. Internal research showed that as much as 71% of parents declared the need for vaccination, although 41% of parents vaccinated their children according to the vaccination calendar. The most frequently mentioned concerns included the possibility of adverse vaccination reactions (22%), the occurrence of autism (7%), and child death (6%). General practitioners had, by far, the greatest impact on the use of protective vaccination in children (73% women and 80% men), although there were cases of discouraging the performance of compulsory vaccinations (41%), and mentioning a doctor (38%) or nurse (3%). *Conclusions*: Modifiable determinants of the negative attitude toward vaccinations are caused mainly by the lack of knowledge. These obstacles in vaccinations can be overcome by improving health education in terms of the vaccination program.

## 1. Introduction

The course of infectious diseases is not always mild, and there is always a risk of serious complications and even death, particularly in younger children. Therefore, the development of preventive vaccinations is considered as one of the greatest achievements of modern medicine. Vaccinations are currently the most effective method of preventing infectious diseases, reducing morbidity and the number of complications and deaths, and allowing complete elimination of the disease. According to the World Health Organization, vaccinations prevent 2–3 million deaths worldwide annually [1,2,3]. Protective vaccinations are recommended for all children. Therefore, their safety is the highest priority and duty of social policy. Before registration, all modern vaccines undergo thorough, meticulous, and reliable safety testing supervised by institutions established in individual countries. In Poland, vaccinations undergo safety testing by the Office for Registration of Medicinal Products, Medical Devices, and Biocidal Products; in Europe—the European Medicines Agency (EMA) and, established by the World Health Organization, the Global Advisory Committee on Vaccine Safety (GACVS); and in the United States—the Vaccine Adverse Event Reporting System [4,5]. Despite the stability of global vaccination coverage, over 19 million children worldwide do not currently receive basic vaccines. Over the past several years, there has been a dramatic drop in the number of vaccinated children worldwide. Recent studies estimate that about one in eight children up to the age of two in the United States are undervaccinated due to their parents’ choice, and most doctors report at least one refusal to vaccinate per month [6,7,8]. Also, in Poland, a growing tendency to avoid vaccination has been observed. In 2011, the parents’ refusal was the reason for not vaccinating about 4700 children, and in 2014, over 12,000 parents refused to vaccinate their children. In 2017, this number increased to over 30,000 children [9,10]. The implementation of the vaccination program and the scope of protection depend on the parents or legal guardians, who decide whether to vaccinate their child or not. Their decision may result in vaccination being delayed or even refused. The increasing number of unvaccinated children is likely due to parents’ concerns about the alleged negative impact of vaccines on health, as well as the availability of unreliable information propagated by the antivaccination movement [11,12,13,14]. Trying to understand the attitudes of parents, their opinion on vaccinations, and the reasons for the decision not to vaccinate their child will allow extensive and appropriately targeted educational activities to be planned and aimed at protecting child’s health through vaccination.

## 2. The Objective of the Work

The study aimed to assess parents’ knowledge, opinions, views, and attitudes about preventive vaccinations, as well as to analyze how healthcare providers impacted the parents’ decision-making.

## 3. Material and Method

### 3.1. Study Design

A population survey was conducted in the Podkarpackie Voivodeship in the Clinical Voivodeship Hospital in Rzeszow, the Healthcare Centre in Jarosław, and Primary Care facilities in Przemysl, Jarosław, and Rzeszow districts in 2018–2019. The main indicator of participation in the study was the age of children, which was between 10 and 14 years, since the majority of compulsory vaccinations were already carried out according to the current vaccination calendar. The Preventive Vaccination Programme (PSO) is the main source of recent information regarding vaccinations in Poland, and it is published annually by the Chief Sanitary Inspector. The Ministry of Health in Poland aims at the implementation of the compulsory vaccination scheme among children using free vaccines funded by the government budget using non–reimbursed combination vaccines and recommended vaccinations against certain infectious diseases included in services paid by caretakers. Doctors and hospital personnel are obligated to provide parents with detailed information regarding compulsory and accessible methods of immunoprophylaxis for their children. The current PSO includes following compulsory vaccines: Tuberculosis (vaccination performed before the child leaves the neonatal unit), diphtheria, pertussis, tetanus (the complete vaccination consists of 4 doses of the vaccine, the first vaccination takes place at the age of 6 to 8 weeks, the second and third at 3–4 months and 5–6 months, and the fourth at 16 to 18 months), measles, rubella, mumps (the first vaccination takes place at the age of 13–14 months, the second at one year), poliomyelitis (the first dose takes place at 3–4 months of age, the second dose at 5–6 months, the third dose at 16–18 months, and an additional dose at the age of 6 years), hepatitis B (the full vaccination cycle includes 3 doses applied in the ages of 0, 1, and 6 months), and infections caused by Haemophilus influenzae B (the full vaccination cycle includes 3 doses applied every 4–8 weeks and an additional dose at the age of 2 years, the first vaccination at 6–8 weeks of age), as well as vaccinations against pneumococcal infections (the full vaccination cycle consists of 3 doses, the first one is applied at the age of 2 months, and doses continue until the age of 5 years). The prevention is complemented by following recommended payable vaccines: Meningococcal infections, chickenpox, rotaviruses, hepatitis A, meningitis, influenza, and human papilloma virus. Parents or legal guardians of the children were invited to participate in the study during their visits to the clinic for healthy or sick children. Each invited person was informed about the purpose of the study. The respondents were given the opportunity to complete an online survey or its paper version.

### 3.2. Sample

The study covered 2300 people residing in the Podkarpackie Voivodeship, 1265 women (55%) and 1035 men (45%). Most of the respondents lived in the city (68%), the remaining persons were rural residents (32%).

### 3.3. Method

The method used in the research was a diagnostic survey, and the technique used was surveying. The research tool was a survey questionnaire. The main part of the questionnaire included questions about parents’ knowledge of current vaccination schedules, vaccine side effects, the presence of contraindications for vaccination, the vaccination qualification procedure, and vaccination safety, as well as preferred sources of information and the role of healthcare professionals in providing information on vaccination and in deciding whether or not a child should be vaccinated. The questionnaire was verified by testing a group of 100 parents in Poland within a month.

### 3.4. Ethical Considerations

The study was approved by the WMU Bioethics Committee (KB 386/2009). Participation in this study was voluntary and anonymous, and respondents were informed of their right to refuse or withdraw from the study at any time. Every respondent was informed of the purpose of the study and the time of completion of the study.

### 3.5. Data Analysis

All data obtained were collected and analyzed using Prism 4.0. Descriptive statistics were used to determine the percentages and 95% confidence interval (CI). Statistical characteristics of continuous variables were presented in the form of arithmetic means, standard deviations, and medians. Statistical characteristics of step and qualitative variables were presented in the form of numerical and percentage distributions using the Student’s *t*-test or Mann–Whitney U test. A correlation was determined using Pearson’s test, while χ^2^ was used for comparison between groups. Significance was assessed at *p* < 0.05. The chi-square test was used to assess the diversity of opinions on vaccination safety in groups. The repeatability of responses to individual questions was assessed using Kappa Cohen statistics. Missing data were excluded from all analyses.

## 4. Results

According to the conducted research, women constituted 55% of participants (95% CI: 51–59), and men constituted 45% (95% CI: 42–51). The average age of men was 44, while for women, it was 41. Almost half of the surveyed women declared secondary education (48%, 95% CI: 41–50), vocational education, (30%, 95% CI: 25–35), or higher education (22%, 95% CI: 19–30). Also, in the case of men, secondary education predominated (43%, 95% CI: 38–47). Vocational education was declared by 37% of respondents (95% CI: 31–41), and higher education was declared by 20% (95% CI: 16–26). Vaccinations carried out negatively correlated with both the age of parents (*p* < 0.0001) and the level of education (*p* < 0.0001). Other descriptive statistics are included in Table 1.

During the assessment of parents’ personal opinions about vaccinations, 71% (95% CI: 69–73) responded favorably, declaring the need to perform them, 18% (95% CI: 13–23) responded indifferently, and 11% (95% CI: 8–18) were against vaccinations. As many as 91% (95% CI: 88–92) of the respondents believed that vaccinations have a significant impact on the child’s health, including as many as 35% (95% CI: 23–37) who believed it is not a positive impact. More than half of the respondents believed that the State should not impose the obligation to vaccinate children (53% of women and 51% of men, 95% CI: 52–60) while, 44% of women and 41% of men (95% CI: 41–48) claimed that vaccinations should be voluntary, and 3% of women and 8% of men (95% CI: 1–19) had no opinion on this subject. A general practitioner had the greatest impact on a parent’s decision to vaccinate, with 73% of women and 80% of men (95% CI: 71–82) following their advice, 44% women and 41% of men claiming that vaccinations should be voluntary, and 3% of women and 8% of men having no opinion on the subject. Other parents declared completely independent decision–making (27% of women and 20% of men, 95% CI: 13–29). According to parents, their independent decision was mostly influenced by research (31%, 95% CI: 23–37), their own feelings and beliefs (26%, 95% CI: 13–41), friends and family experiences (11%, 95% CI: 8–18), their own experience (17%, 95% CI: 11–21), or possible side effects (10%, 95% CI: 8–16). As many as 41% (95% CI: 39–48) of parents admitted that they had encountered the situation of discouraging the performance of compulsory vaccinations, mentioning a doctor (38%, 95% CI: 31–44), friends and family (27%, 95% CI: 22–31), parents of children with vaccine adverse event (7%, 95% CI: 1–14), or a nurse (3%, 95% CI: 1–11). During the assessment of parents’ concerns about vaccinations, more than half (58% of women and 68% of men, 95% CI: 51–69) expressed concern about the allegedly harmful effects of vaccinations on their children’s health, and in particular, of adverse vaccination reactions (22%, 95 % CI: 20–29), autism (7%, 95% CI: 2–13), and child death (6%, 95% CI: 2–13). According to the results of simple analysis, the level of confidence in vaccination safety was associated in a statistically significant way with a negative assessment of the occurrence of vaccination complications. Educated people perceived vaccinations as less safe than those with lower levels of education (*p* = 0.03).

During the analysis of the vaccination system in the study group, it was found that 41% of parents (95% CI: 37–46) vaccinated their children according to the calendar, 41% (95% CI: 37–46) bought combination vaccines to reduce the number of pricks, and 18% (95% CI: 13–25) did not vaccinate their children at all. As many as 40% (95% CI: 39–44) of parents admitted that they refused specific vaccinations (Figure 1). During the assessment of the regularity of vaccinations according to the vaccination calendar, 66% of women and 70% of men (95% CI: 62–72) met the due dates, 26% of women and 24% of men (95% CI: 19–32) did not always meet due dates, and 8% of women and 6% of men (95% CI: 4–7) did not meet the recommended vaccination dates. As a reason for not complying with this obligation, parents mentioned mild infectious diseases of the child (71%, 95% CI: 71–77), chronic infectious diseases (35%, 95% CI: 31–41), the occurrence of convulsions after previous vaccination in a child (12%, 95% CI: 11–17), and 6% mentioned a lack of time (95% CI: 4–9). Among those who did not always or did not comply with vaccination dates, 72% of parents (95% CI: 69–79) vaccinated their child as soon as possible, 22% (95% CI: 19–25) vaccinated their child after receiving a call from the Vaccination Center, and 6% (95% CI: 4–9)vaccinated their child according to their own availability and vaccinations were performed without a schedule. According to the results of the simple analysis, the level of confidence in vaccination safety was statistically significant in relation to the negative assessment of vaccination by professionals. Parents who received a negative opinion on vaccination from a doctor were less likely to decide to vaccinate their child or refused vaccination with specific vaccines (*p* = 0.05).

During the analysis of the use of optional vaccines, it was found that only 28% (95% CI: 19–31) of surveyed parents confirmed the use of optional vaccines. The decision to buy the recommended vaccines was motivated by the fear of falling ill and complications after the disease (84%, 95% CI: 81–89), advertisements (13%, 95% CI: 8–18), or doctor’s recommendation (24%, 95% CI: 19–31). When buying a vaccine for their child, 74% of parents (95% CI: 71–79) took into account the price of the vaccine, 65% (95% CI: 63–72) took into account knowledge about the vaccine, and 58% (95% CI: 54–62) took into account several punctures at one visit and doctor’s recommendations (3%, 95% CI: 1–9). The most frequently mentioned optional vaccines given to children were the pneumococcal (37%, 95% CI: 28–39), rotavirus (25%, 95% CI: 21–32), meningococcal (20%, 95% CI: 13–25), and influenza (3%, 95% CI: 1–5) vaccines. The reason for not purchasing recommended vaccines were financial situation (90%, 95% CI: 88–92), fear of side effects (32%, 95% CI: 31–38), poor information (63%, 95% CI: 53–67), and doubts about the actual effectiveness of these vaccines (13%, 95% CI: 8–18). According to the results of simple analysis, the frequency of nonobligatory vaccinations for motivation was statistically significantly associated with a negative assessment of vaccination costs. People who considered vaccinations expensive were less likely to vaccinate their children with the recommended vaccine (*p* = 0.05).

The study evaluated parents’ opinions about medical personnel and the environment in which vaccinations were carried out. The waiting time was rated positively by 81% (95% CI: 78–84), staff courtesy by 81% (95% CI: 78–84), aesthetics of the rooms by 86% (95% CI: 79–89), office equipment with children’s accessories by 85% (95% CI: 80–88), interest in the child by 85% (95% CI: 80–88), efficiency of performing medical examination by 75% (95% CI: 70–80), efficiency of performing vaccination by 83% (95% CI: 80–88), and competence in providing information on vaccinations and possible complications by 75% (95% CI: 70–80). For the degree of satisfaction with the information received from the doctor about the vaccination schedule and vaccination effectiveness, the results were as follows: 36% of parents (95% CI: 26–41) were very satisfied, 46% were satisfied (95% CI: 42–50), and 18% were dissatisfied (95% CI: 10–22).

The parents’ knowledge of preventive vaccinations was subsequently analyzed. The vast majority of surveyed parents (76% of women and 56% of men, 95% CI: 64–60) declared good knowledge of immunization issues, partial knowledge was declared by 22% of women and 34% of men (95% CI: 19–41), and 2% of women and 10% of men (95% CI: 1–16) did not have knowledge about preventive vaccinations. The information was provided mainly by doctors (53%, 95% CI: 46–59) (Figure 2).

The study showed that 95% of women and 89% of men (95% CI: 84–98) considered vaccines as supplements that stimulate the immune system to raise immunity (85% of women and 86% of men, 95% CI: 84–98). Of the mild reactions that are likely to occur after vaccination, the most often mentioned were fever (27% of women and 30% of men, 95% CI: 24–38), redness (24% of women and 26% of men, 95% CI: 19–30), and edema (23% of women and 24% of men, 95% CI: 19–30) (Figure 3).

As a contraindication for vaccination, the respondents mentioned the child taking an antibiotic (17% of women and 20% of men, 95% CI: 10–22), fever (16% of women and 25% of men, 95% CI: 14–28), and allergic reaction after previous vaccine (73% of women and 51% of men, 95% CI: 47–80). Vaccination qualification should be carried out by the doctor according to 97% of respondents (95% CI: 94–99), while 3% (95% CI: 1–4) believed that it should be done by parents. Most parents (78% of women and 76% of men, 95% CI: 74–79) claimed that unwanted vaccination reactions do not always occur in every child, and that this is an abnormal reaction of the body after vaccination. The most commonly reported symptoms of vaccine adverse event were respiratory symptoms (21%, 95% CI: 10–20), autism (15%, 95% CI: 10–22), and convulsions (23%, 95% CI: 19–30) (Figure 4).

Of the participants, 74% of women and 61% of men (95% CI: 59–79) had heard of combination vaccines. According to them, combination vaccines are new generation vaccines (43% of women and 40% of men, 95% of CI: 39–47), vaccines that reduce the number of punctures (92% of women and 90% of men, 95% CI: 89–97), vaccines immunizing against several infectious diseases simultaneously (68% of women and 54% of men, 95 % CI: 49–70), and safer than traditional vaccines (13% of women and 19% of men, 95% CI: 9–22). In the studied group, there was a very strong positive linear relationship between the source of knowledge and knowing about preventive vaccinations (+0.993), meaning that people acquiring knowledge directly and from reliable sources, i.e., from medical personnel, presented a higher level of knowledge.

The study attempted to analyze the occurrence of vaccine adverse events in the children of the respondents. According to 78% (95% CI: 75–81) of respondents, adverse vaccination reactions occurred. As many as 37% (95% CI: 35–39) did not remember the duration of the adverse vaccination reaction. In 8%, the vaccination reaction lasted 24 h; according to 12% (95% CI: 9–14), it could be observed for 3 days; according to 10% (95% CI: 7–11), it occurred for 7 days; 15% (95% CI: 14–17) replied that the symptoms lasted a few weeks; and 19% (95% CI: 15–20) reported that symptoms lasted a few years. The symptoms mentioned by parents were high fever (23%, 95% CI: 21–25), neurological disorders (14%, 95% CI: 12–16), autism (6%, 95% CI: 5–8), convulsions (20%, 95% CI: 18–21), and developmental disorder (18%, 95% CI: 17–20).

## 5. Discussion

The introduction of common preventive vaccinations remains one of the greatest achievements in public healthcare history. Routine vaccinations during childhood are crucial for the health of individual people, as well as populational health. In Poland, preventive vaccinations are required by law. People temporarily present in Poland are also required, according to the law, to receive compulsory preventive vaccines included in the National Preventive Vaccination Programme. In case a person is not fully able to take legal action, their legal guardian is required to fulfill the obligation. The Chief Sanitary Inspector announces the Preventive Vaccination Programme for a given year in the *Journal of the Ministry of Health*, providing detailed instructions regarding particular vaccinations, the current epidemiological situation, and WHO recommendations. The Programme is published before 31st October in the year preceding the year of the Programme’s implementation. The scheme of obligatory and recommended vaccinations is implemented according to the Programme. The costs of vaccines and performing vaccinations are covered by the part of the government budget controlled by the Minister of Health. The obligatory preventive vaccinations of insured citizens are performed by the healthcare providers that have signed the contract with the National Health Fund. The costs of obligatory preventive vaccinations for uninsured citizens are fully covered by the government budget. Recommended vaccinations are voluntary and performed using fully payable vaccines recommended by the Chief Sanitary Inspector. Due to performing preventive vaccination, the hospital personnel are required to fulfill various law obligations. The performance of the obligatory preventive vaccination needs to be preceded with a medical examination in order to eliminate contraindications for the vaccination. The vaccination cannot be performed more than 24 h after the medical examination date indicated on the certificate. Obligatory vaccinations are performed by doctors, nurses, and midwives. Their duties include informing the patient or their legal guardian of the obligatory vaccination, as well as informing about recommended vaccinations. Moreover, the doctor who suspects or recognizes the occurrence of the adverse effect is obliged to report such event within 24 h to the District Sanitary Inspector [15,16,17,18].

The state of preventive vaccinations of children in Poland is satisfactory. The percentage of vaccinated children is higher than in many other countries of the European Union. It reaches the target values approved by the World Health Organization, the United Nations International Children’s Emergency Fund, and the United Nations Children’s Fund. Despite the majority of children being vaccinated, concerns regarding the safety and effectiveness of vaccines have caused growth in the number of unvaccinated children due to their parents and legal guardians avoiding fulfilling the vaccination obligation [15]. The research conducted by Pieszka (2016) showed that 86% of Polish parents vaccinated their children according to the schedule [16], while the individual research showed that 70.9% of parents expressed the need to vaccinate their children. However, 41.3% of parents vaccinated their children according to the schedule. The percentage of avoiding vaccination varies between countries, but the trend is generally increasing. The research conducted by Heininger U. in Germany showed that 65.1% of respondents refused to vaccinate their children [19]. In the U.S., one in eight children by the age of two is insufficiently vaccinated, and in Australia, 92.2% of children received vaccines in 2012. The percentage of avoiding vaccination has reached the highest level in history—94.78% [6,7,20,21].

The individual research showed that, due to the high cost, only 28.4% of parents chose recommended or combined vaccines. It has been confirmed by the research conducted by Pomian–Osiak, which showed that the cost of combined vaccines was the reason of performing free vaccinations [17]. Also, Kochman and Rudzińska, as well as Pieszka, additionally analyzed the relation between the number of children in the household and buying combined vaccines. It was stated that parents who had one child use combined vaccines more often than parents of three or more children [16,18].

The reason for avoiding the vaccination of children, according to numerous authors, is parents’ concerns. A review of the available literature showed that parents’ beliefs about the possibility of serious vaccination reactions are important factors strongly associated with vaccination fluctuations. Caregivers’ concerns mostly focus on the composition of vaccines, as well as on side effects usually associated with their own negative vaccination experience. The first results of the European project VACSATC, which aimed to track parents’ attitudes to vaccination in several European countries, carried out in 2008–2009, showed a generally positive attitude of parents toward vaccination in childhood vaccination programs [14,22,23]. The most frequently mentioned concerns included the possibility of adverse vaccination reactions (22%), the occurrence of autism (7%), and child death (6%). As shown in the research by Rogalska et al., the main reason for refusal of vaccination of children was the concern about the adverse vaccine effect. The concerns were also related to the simultaneous immunization of children against too many illnesses (34%) and safety of vaccines. (21%) [24]. Similar opinions of parents about an excessive burden of the immune system due to vaccination were described in the research by Heininger et al. [19] and Offit et al. [25]. Most often, younger parents with a lower level of education vaccinate their children according to the calendar, which has been confirmed by data from international literature, emphasizing that the level of parental education also contributed to fluctuations [7]. Opel et al. found that parents with a higher level of education were almost four-times more concerned about vaccine safety than those with a lower level of education [26]. Similarly, Smith et al. found that refusing to accept all childhood vaccines was more common among parents with higher education than those with lower education [27]. Paulussen et al. showed that higher levels of education had a predictive character in case of the negative intention of vaccination [28]. Also, Opstelten et al. found a greater frequency of refusing vaccination among the educated parents [29].

The lack of knowledge about the benefits of vaccination, as well as incorrect information published on the Internet, can influence the parents’ decision to not vaccinate their children. As studies have shown, parents’ fear may be due to the increasing activity of antivaccination movements and their frequent campaigns presenting vaccines as harmful. This has also been confirmed by data published by the Polish State Sanitary Inspectorate, which shows that, for every third unvaccinated child (32%), the parents’ decision was influenced by antivaccination movements [19,20,23,30]. In recent years, many hypotheses have been formulated, of which there is no scientific evidence. They are related to the impact of preventive vaccinations on the occurrence of autoimmune diseases and autism. The hypothesis of the autoaggressive effect of vaccines seems to be similar to the genes of microbial and human proteins, which means that, during vaccination, the immune system mistakenly recognizes and, as a result, attacks its own antigens. The foundations of this theory have remained undocumented. However, vaccination has been shown to protect against exacerbations and infections in patients with autoimmune diseases. Further hypotheses have concerned the impact of vaccines on multiple sclerosis, nephrotic syndrome, type 1 diabetes, or chronic arthritis. The Global Advisory Committee on Vaccine Safety analyzed reports appearing in 2000–2002 regarding the relationship between the occurrence of lymphocytic leukemia after hepatitis B vaccination. These reports have not yet been confirmed [31,32,33]. At the beginning of 1990s, Andrew Wakefield presented a concept of the alleged relationship between the MMR vaccination and autistic disorders. After analyzing available unpublished studies, The Global Advisory Committee on Vaccine Safety (GACVS) clearly stated that this concept was not confirmed. In January 2010, *The Lancet* removed articles by Wakefield from their archive. Evidence supporting this concept was found to be false and manipulated, while Wakefield was found to be a fraudster in a court order [33,34,35]. The following report by Allan Philips focused on the suspected relationship between neurodevelopmental disorders in children using thiomersal, an organic mercury compound showing antiseptic and antifungal properties, which is a preservative used in vaccines in concentrations of 0.001–0.01% [36,37]. In 1999, the Public Health Service and the American Academy of Pediatrics issued a statement calling for the removal of mercury preservatives from all vaccines given to children. Research on the harmfulness of mercury compounds contained in vaccines has also been commissioned. This raised general concern, even though the harmfulness of thimerosal has never been proven. It is now known that there are no contraindications for the use of vaccines containing thimerosal in children and adults, including women who are not pregnant. All studies have shown a lack of relationship between thimerosal and autism [2,5,15,24]. As shown in the research by Facciola et al., the relative acquaintance percentage of people affected by vaccination was 43.3%, but only 13.9% said they knew the person directly. This discovery suggests that the idea of harmful vaccines is present among the public opinion and is easily passed onto others, as indirect knowledge accounted for 29.4% [5]. According to the internal research, as many as 78% of parents claimed that their children had adverse vaccination reactions, but this was due to lack of knowledge about the distinction between vaccination reactions that can occur after each vaccination dose and adverse reactions, which are pathological phenomena.

Parents have a responsibility to make decisions in the best interest of their children. Parents, taking into account medical advice and statistical information on risk and benefits, do not decide to vaccinate their child. The lack of sufficient information on preventive vaccinations often explains the lack of trust in childhood vaccination programs. Usually, it is a lack of knowledge about the etiology of infectious diseases that contributes to the development of so-called antivaccination movements. Various healthcare professionals play a key role in communication with parents. According to Facciola et al., only 18% of doctors favored recommended vaccinations, while 47.5% were opposed to vaccines [5]. Internal research has not confirmed this tendency, as General Practitioners have, by far, the greatest impact on the use of protective vaccination in children (73% women and 80% men), although there were cases of discouraging the performance of compulsory vaccinations (41%), and mentioning a doctor (38%) and nurse (3%). Similar results were obtained by Heininger U., which showed that GPs have the most significant role in educating parents regarding vaccinations. A GP, as the most available and reliable source of information, is a person who has the greatest impact on the parents’ decision. Among 6025 respondents, 5722 (95.0%) mentioned their GP as the most important source of information on immunization, before leaflets (48.0%), health magazines (44.7%), and the Internet (38.7%) [19].

Insufficient contact with doctors and nurses, lack of reliable information from specialists regarding safety, mechanisms and effectivity of vaccinations, and negative attitude are important factors influencing parents’ attitude toward children vaccination, and they can even become the greatest obstacle in vaccination implementation [2,24,38]. Although a randomized study by Wilson revealed that neither evidence-based teaching nor presentations of polio survivors changed the chiropractics’ perception of vaccination, it is assumed that education and discussion with practitioners are ways to increase the acceptance of vaccinations [4,39,40,41,42,43]. Therefore, healthcare professionals, as internal research has also shown, should provide complete information on the risks and benefits of immunization and targeted diseases, as well as information on the effectiveness and risk of alternative methods, including the refusal of vaccination. Information should be presented in a way that supports conscious decision–making, which provides parents with the necessary basis to make accurate decisions [4,44,45,46]. Highly educated parents should choose such strategy to be their priority. Various research, including individual research, has shown that educated people more rarely perceived vaccinations as safe, compared to respondents with lover levels of education (*p* = 0.03). More effective cooperation with the society is required, which can be achieved by providing reliable information and discussions. In this field, the educational role of medical personnel is of the utmost importance. Due to their direct contact with parents, medical workers are oriented in the current situation and have the possibility, as well as the obligation, of educating the concerned parents and caretakers [9].

## 6. Conclusions

Modifiable determinants of the negative attitude toward vaccinations are caused mainly by the lack of knowledge. These obstacles in vaccinations can be overcome by improving health education in terms of the vaccination program.The cost of combined and recommended vaccines is a significant financial obstacle for Polish parents. That should convince Polish legislators to take into consideration at least partial reimbursement of combined and recommended vaccines.Regular monitoring of parents’ attitudes toward the vaccination program will allow for the adjustment of educational programs to current needs.

## Figures and Tables

**Figure 1 ijerph-17-03694-f001:**
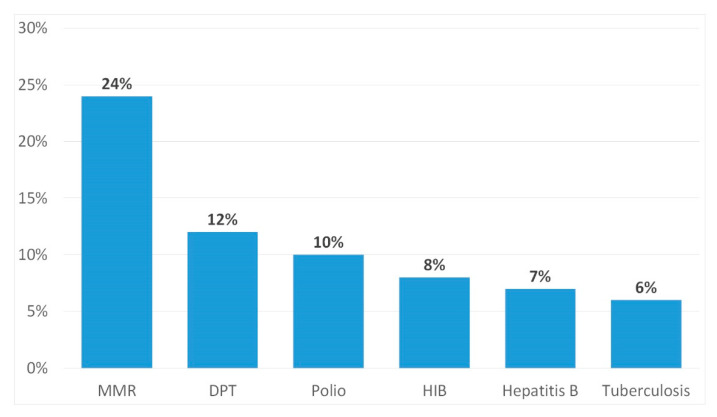
Refusal to vaccinate children.

**Figure 2 ijerph-17-03694-f002:**
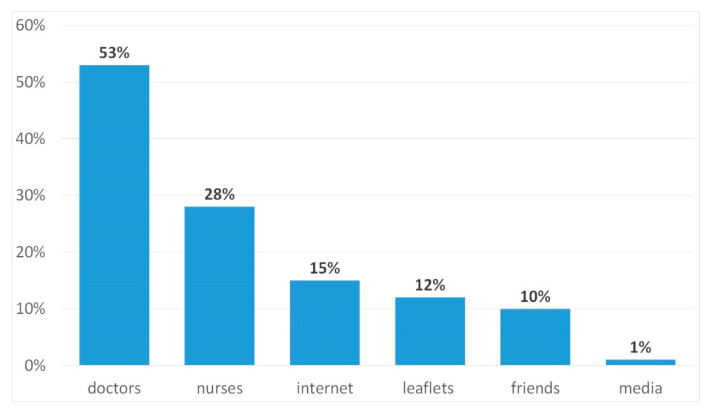
Sources of information about vaccinations.

**Figure 3 ijerph-17-03694-f003:**
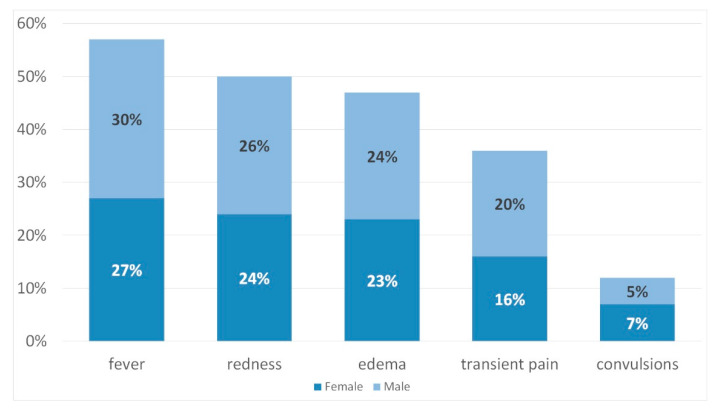
Parents’ knowledge of possible vaccination reactions.

**Figure 4 ijerph-17-03694-f004:**
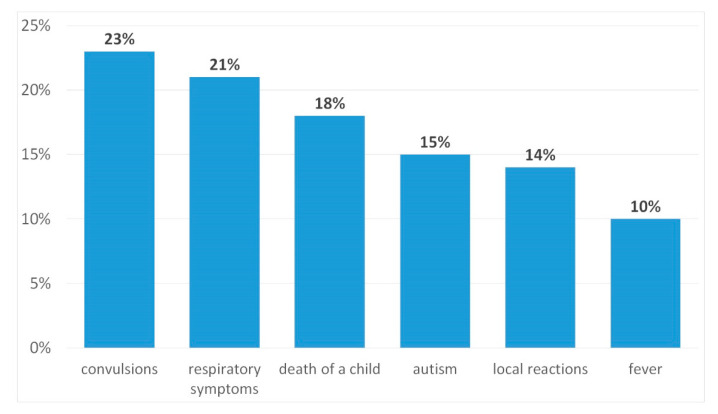
Parents’ knowledge of possible vaccine adverse event.

**Table 1 ijerph-17-03694-t001:** Descriptive statistics of the examined group of parents.

	Total
**Sex**	*n* = 2300
women (N/%)	1265/55%
men (N/%)	1035/45%
**The age of Women**	
± standard deviation	41.2 ± 7.01
scope	[23; 55]
median	41
95%CI	[39.8; 41.8]
**The age of Men**	
± standard deviation	43.8 ± 4.9
scope	[27; 52]
median	44
95%CI	[43.3; 44.0]
**Place of Residence**	
city (N/%)	1564/68%
village (N/%)	736/32%
**Financial Situation**	
very good (N/%)	184/8%
good (N/%)	1058/46%
average (N/%)	1012/44%
bed (N/%)	46/2%
**Number of Children Owned**	
one child (N/%)	1265/55%
two children (N/%)	874/38%
three children (N/%)	161/7%
**Women’s Education**	
higher education (N/%)	278/22%
secondary education (N/%)	607/48%
vocational education (N/%)	380/30%
**Men’s Education**	
higher education (N/%)	207/20%
secondary education (N/%)	445/43%
vocational education (N/%)	383/37%

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
