# Peer review of "Opinions and Knowledge of Parents Regarding Preventive Vaccinations of Children and Causes of Reluctance toward Preventive Vaccinations"

_ijerph, 2020, doi:10.3390/ijerph17103694_

Round 1

Reviewer 1 Report

This is a very interesting and valuable study that documents trends of vaccinations in children in Poland in current times and investigates reasons for their decline including parents' attitudes and knowledge about vaccines. Overall, this is a good study and is easy to understand, but the manuscript still requires editing by a Native English speaker for grammar. In addition, some points are listed below that should be addressed:

While the results are good, they need to be presented in a manner that is easier to understand.  For example, the results in the abstract need to be reworded - the wording is currently confusing and not all of the important findings are included. The "Abstract conclusions" are actually part of your results and not a conclusion. Be consistent with the format you use to present your data, i.e. "(48%, 95% CI:41.3-50.1)". I would also strongly encourage the authors to include charts and/or graphs to help visualize the results. Perhaps the questionnaire could be included as supplementary information.

In the introduction it would be helpful to cite studies from other countries that have shown similar trends in reduced vaccination rates in children.

Study design - were all clinics in this region invited to participate in the study? how many of them joined the study? is the study area rural or in the city?

Discussion - it would be helpful to indicate which vaccines are covered by healthcare and which need to be paid for by the parents

Minor comments:

Line 18 - change "decide to vaccine or not their child" to "decide to vaccine their child or not"

Line 18 - change "parents 'decisions" to "parents' decisions"

Line 20 - please include where the population survey was conducted, i.e. in Poland, Europe etc

Line 23 - change "is" to "was"

Line 24 - change "women -" to "in women it was"

Line 26 - change "believe" to "believed"

Line 29 - add "knowledge about the benefits of vaccinations"Line 36 - change "and they pose..." to "and there is always a risk of"

Line 39 - delete "they are the most effective preventive action" - repetitive

Line 40 - add "and allowing"

Line 43 - change to "Before registration all modern vaccines..."

Line 47 - add "the Global"

Line 47 - place "EMA" in brackets

Line 48 - place "GACVS" in brackets

Line 55 - delete "-"

Line 57 - change to "to vaccinate their child or not"

Line 62-64 - change to "will allow extensive and appropriately targeted educational activities to be planned and aimed at protecting child's health through vaccination"

Line 67 - change to "as well as to analyze how healthcare providers impacted in the parents' decision making"

Line 70 - add "in Poland"

Line 81 - change to "and the technique used was surveying"

Line 87 - if correct, change to "verified before the beginning of the study by..."

Line 96 - add "and medians"

Line 105 - change to "women was 41.2"

Line 108 - change to "predominated"

Line 111 - change to "lived"

Line 115 - change to "had one child", "had two children", "had three children"

Line 119 - change "was" to "were"
Line 119 - change to "believed"

Line 120 - change to "including as many as 35%"

Line 121 - change to "believed there is no positive impact", change to "believed"

Line 123 - change to "claimed"

Line 124 - change to "had no opinion"

Line 124 - change to "A general practitioner had the greatest impact on a parent's decision to vaccinate, with 73% women and 80% of men following their advice, 44% women and 41% men claiming that vaccinations should be voluntary, and 3% womena dn 8% men having no opinion on the subject."

Line 129 - change to "friends'"

Line 132 - change to "friends and family"

Line 133 - change to "and nurse"

Line 136 - delete "-"

Line 137 - change to "and child death"

Line 146 - change "against polio" to "the polio vaccine", change "against Hepb" to "the hepatitis B vaccine", change "against TB" to "the BCG vaccine"

Line 267 - delete "including those operating in life"

Line 277 - please include the year that this report came out.

Line 286 - do you mean women who ARE pregnant?

Author Response

Dear Reviewer,

Thank you very much for the comments of the reviewers. After analyzing all the comments, I made the following changes:

  1. The work was checked and corrected by a native speaker.
  2. Changes were made to the summary, the results and conclusions were rewritten.
  3. Consistency was introduced in the presentation of data. The results are presented in whole number.
  4. In the results section (line 134-148) a table with the characteristics of the research group has been added.
  5. In the results section (line 149-263) 4 results figures have been added.
  6. The discussion section (line 264-454) compared the results with other similar studies described in the literature. Similarities and differences are highlighted.
  7. The material and method section (line 75-97) and the discussion section (line 265-299) describe the current preventive vaccination program for children in Poland.
  8. The applications were analyzed and changed according to the results (lines 456-469).
  9. In the material and method section (lines 76-79) the facilities where tests were performed were added.
  10. In the material and method section (line 108-110) information about the residence of the study group was added.

I hope that the changes made are satisfactory and this will allow publication. I am asking you to take into account the positive comments of the reviewers that this is an interesting study and a good study.

Sincerely

Author

Reviewer 2 Report

The manuscript entitled "Assessment and knowledge of parents regarding preventive vaccinations of children and entering into contact with preventive vaccinations" to assess parents' knowledge, attitudes, and beliefs about vaccines since their affect the success of the vaccine program and the tax of immunization of the children. The survey was conducted between 2018 and 2019. The authors present interesting results regarding the profile of the parents regarding their opinion about vaccination. There are some points that can aid to improve the visibility of the manuscript. 1 - Results section Please, it is suitable to add a table(s) for the results. Please, it will be more clear for the reader and will improve the visibility of the manuscript. 2 - Discussion - Please, compare the results with the other similar studies reported in the literature. Highlighting the similarities and differences. 3 - Methods section - study design subsection "The main indicator of participation in the study was the age of children - between 10 and 14 years since the 72 majorities of compulsory vaccinations are already carried out according to the current vaccination calendar" Please give more details about the vaccines. 4 - Results section - Additionally, to compare some results some plots will increase the visibility of the manuscript. 5 - Conclusion section "1 - People acquiring knowledge directly and from reliable sources, from medical personnel, presented a higher level of knowledge." "3 - Educated people perceived vaccinations as less safe than those with lower levels of education." Please the authors must analyze and rewrite the conclusion section, points 1 and 3 didn't corroborate between them. Therefore these points must be clarified succinctly in this section. 6 - Discussion section - page 6 line 250 until page 7 line 317 Please this paragraph is too extended, I think that the authors will follow the suggestions 1 and 4, it will possible to rewrite all results and discussion sections and this paragraph will be divided being more readable.

Author Response

Dear Reviewer,

Thank you very much for the comments of the reviewers. After analyzing all the comments, I made the following changes:

  1. In the results section (line 134-148) a table with the characteristics of the research group has been added.
  2. The discussion section (line 264-454) compared the results with other similar studies described in the literature. Similarities and differences are highlighted.
  3. The material and method section (line 75-97) and the discussion section (line 265-299) describe the current preventive vaccination program for children in Poland.
  4. In the results section (line 149-263) 4 figures with results have been added.
  5. The applications were reviewed and amended according to the results.
  6. The text has been shortened, rewritten, which will probably facilitate reading to the reader.

I hope that the changes made are satisfactory and this will allow publication. I am asking you to take into account the positive comments of the reviewers that this is an interesting study and a good study.

Sincerely

Author